# Genomic signatures reveal DNA damage response deficiency in colorectal cancer brain metastases

Jing Sun[1,13], Cheng Wang[2,3,13], Yi Zhang[4], Lingyan Xu[1], Weijia Fang[5], Yuping Zhu[6], Yi Zheng[5], Xiaofeng Chen[1], Xiju Xie[7], Xinhua Hu[8], Weidong Hu[9], Jingyu Zheng[10], Ping Li[1], Jian Yu[11], Zhu Mei[1,12], Xiaomin Cai[1], Biao Wang[1], Zhibin Hu[2], Yongqian Shu[1,14], Hongbing Shen[2,14] & Yanhong Gu[1,14]

Brain metastases (BM) of colorectal cancer (CRC) are rare but lethal, and an understanding of their genomic landscape is lacking. We conduct an analysis of whole-exome sequencing (WES) and whole-genome sequencing (WGS) data on 19 trios of patient-matched BMs, primary CRC tumors, and adjacent normal tissue. Compared with primary CRC, BM exhibits elevated mutational signatures of homologous recombination deficiency (HRD) and mismatch repair deficiency (MMRD). Further analysis reveals two DNA damage response (DDR) signatures could emerge early and are enhanced in BM tissues but are eliminated eventually in matched primary CRC tissues. BM-specific mutations in DDR genes and elevated microsatellite instability (MSI) levels support the importance of DDR in the brain metastasis of CRC. We also identify BM-related genes (e.g., *SCN7A*, *SCN5A*, *SCN2A*, *IKZF1*, and *PDZRN4*) that carry frequent BM-specific mutations. These results provide a better understanding of the BM mutational landscape and insights into treatment.

[1] Department of Oncology, The First Affiliated Hospital of Nanjing Medical University, Nanjing 210029, China. [2] Department of Epidemiology and Biostatistics, School of Public Health; Jiangsu Key Lab of Cancer Biomarkers, Prevention and Treatment, Jiangsu Collaborative Innovation Center for Cancer Personalized Medicine, Nanjing Medical University, Nanjing 211116, China. [3] Department of Bioinformatics, School of Basic Medical Sciences, Nanjing Medical University, Nanjing 211116, China. [4] Department of General Surgery, The First Affiliated Hospital of Nanjing University of Chinese Medicine, Nanjing 210029, China. [5] Cancer Biotherapy Center, The First Affiliated Hospital, School of Medicine, Zhejiang University, Hangzhou 310006, China. [6] Department of Colorectal Surgery, The Zhejiang Cancer Hospital, Hangzhou 310022, China. [7] Department of General Surgery, The First Affiliated Hospital of Nanjing Medical University, Nanjing 210029, China. [8] Department of neurosurgery, The Affiliated Brain Hospital of Nanjing Medical University, Nanjing 210029, China. [9] Department of Neurology, The Second Affiliated Hospital of Soochow University, Suzhou 215004, China. [10] Department of Pathology, The Affiliated Drum Tower Hospital, Nanjing University Medical School, Nanjing 210008, China. [11] Department of Pathology and Radiation Oncology, University of Pittsburgh School of Medicine, Pittsburgh, PA 15213, USA. [12] Department of Oncology, Sir Run Run Hospital, Nanjing Medical University, Nanjing 211116, China. [13]These authors contributed equally: Jing Sun, Cheng Wang. [14]These authors jointly supervised this work: Yanhong Gu, Hongbing Shen, Yongqian Shu. Presented in part at the Gastrointestinal cancer symposium of the America Society of Clinical Oncology, San Francisco CA, 18–20 Jan 2018. Correspondence and requests for materials should be addressed to Y.G (email: guyanhong@njmu.edu.cn), H.S. (email: hbshen@njmu.edu.cn) or Y.S. (email: shuyongqian@njmu.edu.cn)

Colorectal cancer (CRC) is the third most common cancer in males and second most common cancer in females[1]. Distant metastases occur frequently in CRC patients. Brain metastasis (BM) is much less frequent from CRC than from other common cancers such as lung, liver, breast or kidney cancers. The incidence of BM in CRC is estimated to range from 1 to 3%[2] and has been increasing in recent decades. The prognosis of BM patients is poor, with a median survival time of 3–6 months upon diagnosis and no effective therapy[3].

The treatment of metastatic CRC has changed dramatically in recent years with targeted therapies. However, treatments for CRC BM patients remain largely based on primary tumors. The genomic landscapes of CRC have been extensively studied and have identified several actionable therapeutic targets[4]. A recent sequencing study on multiregion samples from 17 CRC patients reported that most local and distant metastases arise from independent subclones within the primary tumor[5], indicating genetic divergence and heterogeneity of metastatic diseases. Several BMs from various tumor types were studied by whole-exome sequencing (WES), supporting the notion that BM is genetically distinct from the regional lymph nodes and extracranial metastasis[6]. A higher mutation rate of KRAS/NRAS and PIK3CA genes was reported in brain metastasis tissues[7,8]. However, it remains largely unknown whether distinct genomic features exist in BM and primary CRC.

In this study, we conduct WES and whole-genome sequencing (WGS) of biopsy samples from trios of patient-matched brain metastases, primary CRC tissues and adjacent normal samples (WES: 42 tissues from 11 patients; public WES: 13 tissues from 4 patients; WGS: 24 tissues from 8 patients) to (1) document genomic signatures during the evolution of CRC BM; (2) identify clinically actionable targets for the BM treatment; and (3) identify potential drivers in CRC BM.

## Results

**Mutational signatures of brain metastases and matched primary tumors.** We conducted WES on 42 tissues from 11 CRC patients with brain metastases and WGS on 24 tissues from 8 CRC patients with brain metastases. The clinical characteristics of the 19 patients are summarized in Supplementary Table 1. We also included 13 previously published WES data sets in the same sample trio format, derived from 4 patients[6] to increase the power of our analysis. In total, 21,181 (WES) and 277,617 (WGS) unique somatic substitutions were identified in all primary and metastatic tumor tissues by comparing them with the matched adjacent normal samples (Supplementary Table 2). Only 31.3% (1416/4519, WES) and 28.9% (45264/157377, WGS) of the mutations in BMs were identified in primary CRC, indicating the evolution and selection of mutations during metastasis. We classified all point substitutions into six groups according to the direction of the mutation. As expected, the proportions of all six mutation groups were significantly different across BMs and matched primary tissues: C > T mutations were reduced in BM tissues compared with those in primary tissues, while the reverse was true of the other mutation types (Fig. 1a).

To further determine the temporal dynamics of the BM genomic landscapes, we analyzed mutational signatures in BM and primary CRC using deconstructSigs, which accurately reconstructs the mutational profiles of each sample based on a predefined mutational spectrum of 30 COSMIC signatures (Supplementary Data 1). We included the mutation profiles of 489 TCGA CRCs in the analysis for comparison. Signatures 1, 3 and 6, which have been linked to aging, HRD and MMRD, respectively[9], were identified as dominant in either BM or primary tissues (Fig. 1b and Supplementary Figs. 1 and 2).

Interestingly, the proportion of the aging-related mutational signature was almost 80% in our cohort of the primary metastatic CRC samples but <50% in the TCGA CRC samples (Fig. 1b and Supplementary Table 3). This signature was decreased dramatically in BMs, while HRD and MMRD signatures were significantly elevated in BMs (Wilcoxon rank sum test $P < 0.05$, Fig. 1c). To validate our results, we further performed RNA-Seq on BMs and primary tissues from only two additional CRC patients. Differential expression analysis revealed that several mismatch repair or homologous recombination genes (e.g., EXO1 and BLM) were significantly downregulated in BMs (Supplementary Data 2). GSEA analysis also suggested that MMR and HR gene sets were significantly negatively enriched in brain metastasis tissues (Supplementary Fig. 3).

HRD mutations were rare in the TCGA or primary CRC patients (Fig. 1b). MMRD featuring MSI is commonly seen in hereditary nonpolyposis colorectal cancer (HNPCC) or Lynch syndrome patients[10] but is seldom reported in the primary sites of BM patients. Interestingly, Signature 17 commonly occurs in breast cancer[11] and metastasis tissues[12]. It was recently found to emerge in the genome after the loss of MMR[13]. In our data, the proportion of Signature 17 was significantly correlated with the mutation burden (r = 0.58, Spearman correlation coefficients $P = 1.72 \times 10^{-5}$, Supplementary Fig. 4). In contrast to brain metastasis, the liver metastasis of CRC did not show the elevation of the signatures (Supplementary Fig. 5).

To better investigate the dynamics of BM mutational signatures, we separated all mutations into three categories: mutations found only in BM tissues, mutations shared by BMs and primary CRC, and mutations only in primary CRC (Fig. 1d). The shared mutations represent early and ancestry mutations. HRD and MMRD signatures are enhanced in BM, suggesting the unstable feature is likely to be an early event. By contrast, the primary-only mutations were dominated by Signature 1 (Fig. 1d) in whole-exome data and are associated with increasing age[9]. Thus, CRC BM appears to be enriched for greater genomic instability and DNA repair deficiency, which evolves early and independently from the gradual accumulation of abundant aging-related mutations at the primary site.

**Evolution of somatic mutations in DNA damage response genes.** DDR is critical for maintaining genome stability. Deregulated DDR is a hallmark of cancer and is caused by mutations in checkpoints and DNA damage-sensing and -repair proteins involved in homologous recombination and mismatch repair pathways. Several drugs targeting abnormal DDR in cancers have been approved by the US Food and Drug Administration (FDA) or are under clinical investigation[14]. Functional mutations in DDR genes have been shown to be responsible for the characteristic mutational signatures in tumors[9,14]. Thus, we systematically evaluated the CCF of all nonsynonymous mutations in 353 expert-curated human DDR gene sets[14] to further define the evolution of these mutations.

Fifty-four nonsynonymous mutations in 35 DDR genes were identified in our 19 BM tissues (Supplementary Data 3). TP53 was the gene with the highest mutation rate (74%, 14/19), but almost all TP53 mutations were shared between BMs and primary tissues (Fig. 2) and occurred in the founding clones of each patient (Supplementary Fig. 6), suggesting that TP53 mutations are drivers and occur prior to BM. Forty-six percent of the mutations (25/43) occurred only or mainly in BMs. Interestingly, we observed a biallelic loss-of-function BRCA1 mutation in the BM of patient BM001, and the mutation was heterozygous in the matched primary tissues. We identified two BM-only heterozygous BRCA2 mutations in patient BM007 (Fig. 2 and

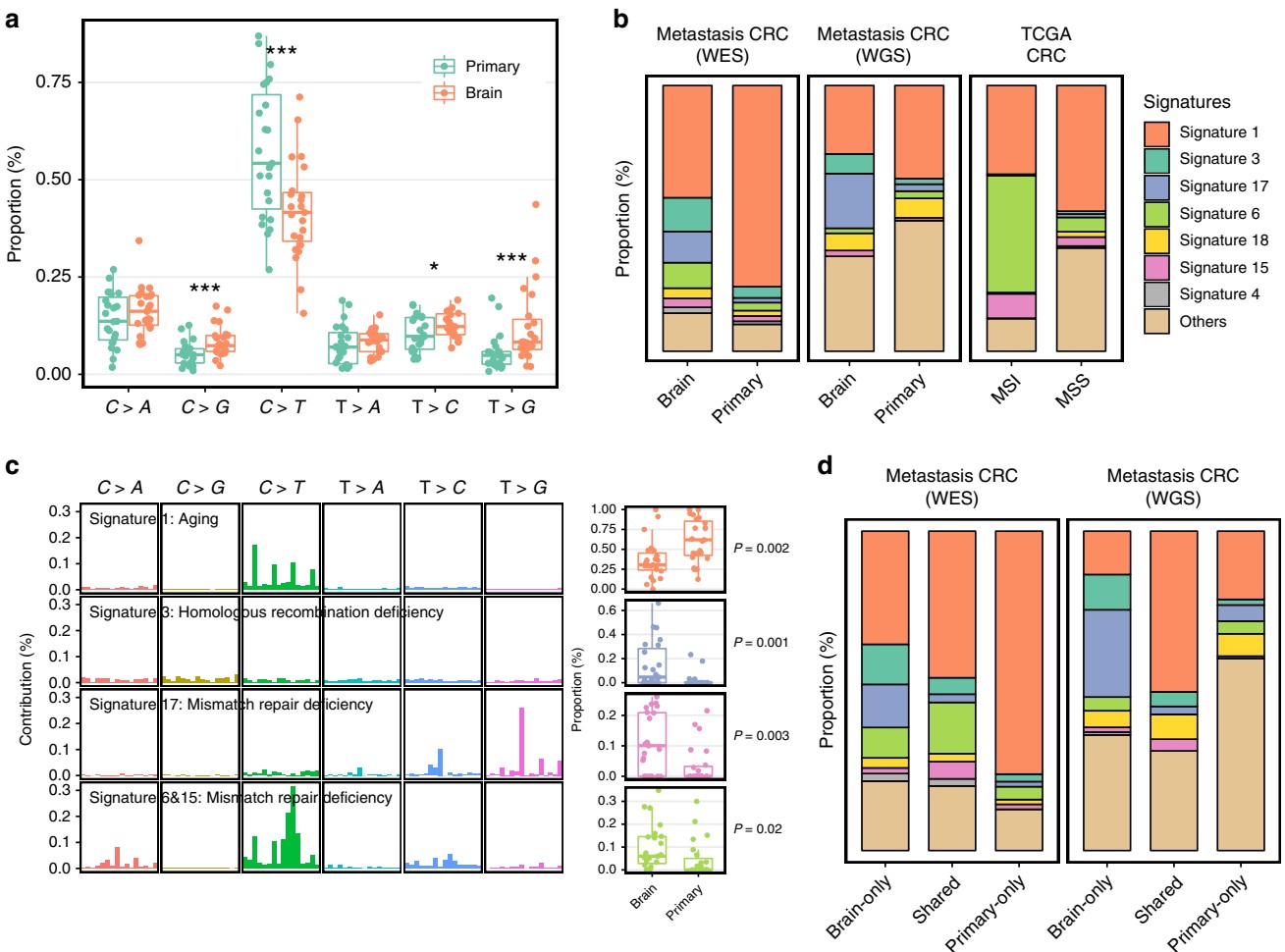

**Fig. 1** Divergent mutational features in brain metastases and matched primary tumors of CRC patients. **a** Six mutational subtypes in BM and primary tissues. The BM tissues are presented in orange, and the primary tissues are presented in cyan. **b** Major mutational signatures in our patients and TCGA CRC patients. We reconstructed the proportion of mutational signatures of each sample based on a predefined mutational spectrum of 30 COSMIC signatures. The proportion of the top seven signatures in BM is presented. **c** HRD and MMRD signatures were significantly elevated in BM tissues compared with those in primary tissues. Four mutational signatures with a significant difference between BM and primary tissues are shown in the figure. The mutation characteristics are presented at the left side and the box plot of the proportion of each signature is presented at the right side. The box plot displays the first and third quartiles (top and bottom of the boxes), median (band inside the boxes), and lowest and highest point within 1.5 times the interquartile range of the lower and higher quartile (whiskers). **d** All mutations were classified into three groups—brain only, shared and primary only—and mutational signatures were extracted in each group

Supplementary Fig. 7A, B). Both patients carried a higher proportion of HRD mutations in the BM than in the primary tissues (Supplementary Fig. 7C), and clonal analysis suggested that they were the founding mutations of brain metastasis (Supplementary Fig. 6). These mutations may contribute to the elevated DNA damage related-mutations in the metastasis. Patients BM013 and BM014 carried BM-only mutations on *RAD51B* and *PAXIP1*, respectively, which were the important participants of homologous recombination. BM-only mutations also occurred on the nonhomologous end joining (NHEJ) gene *XRCC4* (BM018). The patients also had an elevated level of the HRD signature (Fig. 2 and Supplementary Fig. 7C). Two other patients were found to have distinct patterns of HRD signatures (proportion difference > 0.1) in BMs and primary tissues (Supplementary Fig. 7C). Patient BM002 carried BM-only mutations in the translesion synthesis (TLS) gene *SHPRH* and chromosome segregation gene *PDS5B*. Patient BM004 carried BM-founding mutations in the *POLB* gene (Supplementary Fig. 6), which is involved base excision repair (BER) and nonhomologous end joining (NHEJ). Similarly, patient BM010

carried BM-founding mutations in checkpoint factor *CHEK1* and homologous recombination gene *SLX4* (Supplementary Fig. 6). Because DDR proteins often interact with one another to form a complex network[14], these BM-specific or enriched DDR mutations likely contribute to the elevated HRD signature.

**Elevated MSI level and immunological infiltration in BM tissues.** To further validate the elevated MMRD mutational signature in BM, we measured the somatic MSI index using an orthogonal bioinformatics approach with MSIsensor. BM showed significantly elevated MSI levels compared with matched primary CRC (Wilcoxon rank sum test $P = 0.002$) (Fig. 3a). Among 16 patients with microsatellite-stable (MSI-negative, MSI score < 3) primary CRC, 4 patients (25%) developed MSI-positive BMs. Interestingly, liver metastases from 10 other CRC patients did not show greatly elevated MSI levels (Fig. 3a).

The MMRD mutational signature is commonly associated with increased lymphocyte infiltration, a favorable target for immunological therapy[15]. Thus, we analyzed TILs in BM using

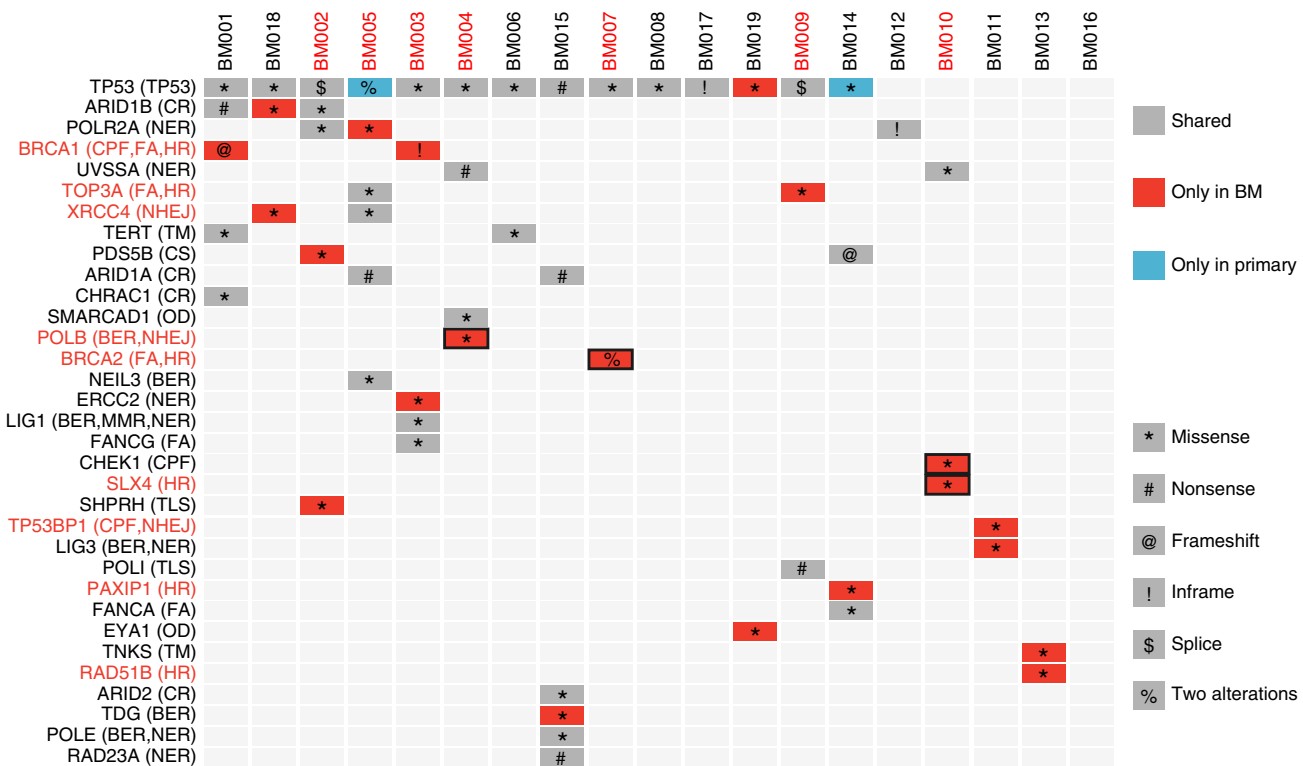

**Fig. 2** Mutational spectrum of DDR genes in brain metastases and matched primary tumors of CRC patients. Mutations occurring only in BM tissues are presented in red, and mutations occurring only in primary tissues are presented in blue; shared mutations are presented in grey. Genes involved in homologous recombination and nonhomologous end joining are marked in red. Bold rectangle indicates that the mutations appear in the founding clones

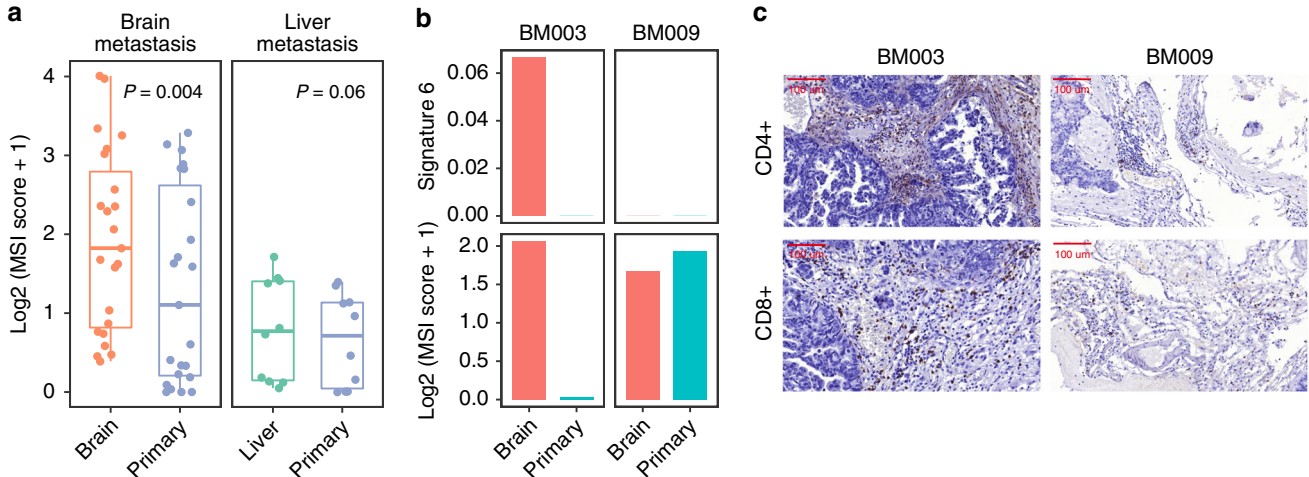

**Fig. 3** MSI statuses and tumor-infiltrating lymphocytes in brain metastases and matched primary tumors of CRC patients. **a** MSI scores were significantly higher in BM tissues than in primary tissues but were not elevated in liver metastatic tissues. The box plot displays the first and third quartiles (top and bottom of the boxes), median (band inside the boxes), and lowest and highest point within 1.5 times the interquartile range of the lower and higher quartile (whiskers). **b**, **c** Patient BM003 showed a consistent elevation of signature 6 mutations and the MSI score in BM tissues, whereas patient BM009 did not. In the brain tissues of patient BM003, there was a high degree of infiltration by CD4+ and CD8+ T cells

immunohistochemistry. BM003 was one of the four patients who developed the BM-specific MMRD mutational signature and MSI with a high abundance of CD4$^+$ and CD8$^+$ infiltrating lymphocytes, while the matched primary tumor carried a low level of MMRD mutations and a low MSI score (Fig. 3b, c). By contrast, patient BM009, who had a similar level of MMRD in BMs and primary tissues, showed a low level of immunological infiltration.

**Potential driver genes in brain metastases**. To discovery potential CRC BM drivers, we first excluded known passengers (*TTN* and olfactory factor genes) and identified 38 genes that had at least 3 mutations in BM. Five of them (*TP53*, *APC*, *KRAS*, *FBXW7*, and *MUC17*) were identified as significantly mutated genes in CRC in a previous study[16] (Fig. 4a). Next, we compared the mutation patterns between our CRC BM cohort and TCGA CRC cohort. Twelve genes (*EYS*, *MUC17*, *PDZRN4*, *DLX5*,

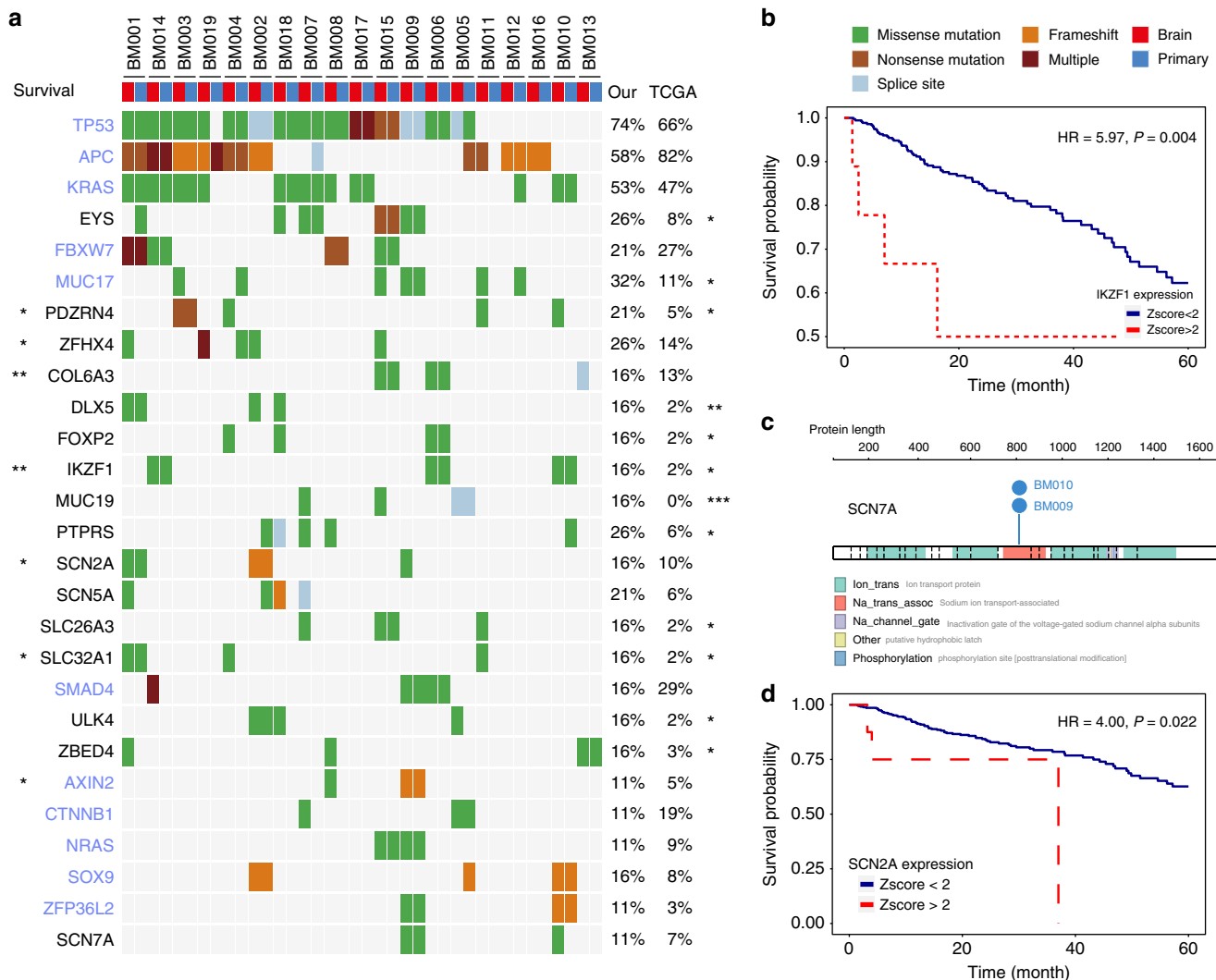

**Fig. 4** Potential driver genes related to BM. **a** Twenty-seven genes included (1) genes with more than 3 mutations in BM tissues; (2) genes with a significantly different mutation rate between our patients and TCGA patients; (3) genes with expression or mutations associated with the overall survival of patients in TCGA; (4) genes reported as significantly mutated genes in a previous study and with at least 2 mutations in BM tissues; (5) genes encoding voltage-gated sodium channel proteins with at least 2 mutations in BM tissues. Significantly mutated genes identified in a previous study are colored in light blue. The mutation rate in our cohort and TCGA data are listed on the right side and compared. The $P$ value of the Cox regression model corrected for age, sex, and stage in survival analysis is listed on the left side. *$P < 0.05$; **$P < 0.01$. **b** In the TCGA data, the expression of *IKZF1* was significantly associated with poor survival among CRC patients. **c** A recurrent mutation in *SCN7A* was identified in the BM tissues of two patients. **d** In the TCGA data, the expression of SCN2A was significantly associated with poor survival among CRC patients

*FOXP2, IKZF1, MUC19, PTPRS, SLC26A3, SLC32A1, ULK4,* and *ZBED4*) showed significantly elevated mutation rates in our cohort compared with those in the TCGA cohort or matched TCGA cohort corrected for age, sex and stage at diagnosis (Fig. 4a, Supplementary Fig. 8). Notably, *MUC19* occurred only in BM tissues. *SLC32A1* was specifically expressed in normal brain tissues (Supplementary Fig. 9), while *PDZRN4* was mainly expressed in normal colon and nerves (Supplementary Fig. 10). Three in four *PDZRN4* mutations were observed only in BM and not previously observed in the COSMIC database, while a nonsense mutation (p.R965*) showed an elevated cancer cell fraction in BM tissues (100% *vs.* 43%) (Supplementary Fig. 11A). Nonsense and missense mutations at p.R965 in stomach adenocarcinoma and other cancers were found in the COSMIC database. *IKZF1* mutations occurred more frequently in our patients, and its expression was significantly associated with poor survival (Fig. 4b). *IKZF1* is a hematologic system specific expressed gene and was reported to play a critical role in lymphocyte

differentiation and development[17,18]. Recently, the MKSCC study also reported that *IKZF1* is a significantly recurrent mutated gene in MSS CRC cancer, indicating the importance of *IKZF1* in colorectal cancer[19]. Interestingly, though *IKZF1* was mainly expressed in lymphocytes and was downregulated in CRC tumor tissues (Supplementary Fig. 12A), single cell RNA-Seq data[20] suggested that it was also highly expressed in 4.0% of tumor cells (FPKM > 5) and in none of the normal mucosa cells (Fisher's exact test $P = 0.009$, Supplementary Fig. 12B). The results suggested that *IKZF1* is expressed in both tumor cells and lymphocytes. Thus, the mechanisms of the *IKZF1* role in CRC prognosis could be complicated and may involve tumor cells as well as infiltrating lymphocytes. Further well-designed experiments are required for the investigation of the underlying mechanism of *IKZF1*. Among 118 significantly mutated genes identified previously in CRC[16], only 11 genes carried at least two mutations, and only *MUC17* exhibited a significantly elevated mutation rate in comparison to TCGA data (Fig. 4a, Supplementary Fig. 8),

suggesting that the development of brain metastasis is unlikely driven by known driver genes.

We also identified two patients carrying a recurrent missense mutation in the voltage-gated sodium channel protein *SCN7A* (Fig. 4c). For patient BM010, the mutation occurred only in brain tissue and not in primary CRC tissue (Fig. 4a). For patient BM009, although the mutation occurred in both BM and primary CRC, the CCF was much higher in BM than in CRC tissues (100 vs. 16%). More importantly, both mutations belonged to the founding clones of BM009 and BM010 (Supplementary Fig. 6). The recurrent mutations were validated by Sanger sequencing after TA cloning (Supplementary Fig. 13). This mutation (p. L805P/R) was not found in either the COSMIC or TCGA data.

Another voltage-gated sodium channel protein, *SCN5A*, was also identified as a frequently mutated gene in our BM patients. Three of the four mutations were BM specific (Fig. 4a). We also found that *SCN2A* carried a frequent mutation in BM patients (Fig. 4a). The mutation was located in the sodium ion transport-associated domain, which was also impaired in *SCN7A*. The other two *SCN2A* mutations were also located in important domains of the voltage-gated $Na^+$ ion channel (Supplementary Fig. 11B). In the TCGA data, higher *SCN2A* expression was significantly associated with poorer survival among CRC patients (Fig. 4d).

## Discussion

Brain metastasis is rare in CRC patients, and its evolutionary process at the genomic level remain largely unknown. Recent WES analyses revealed independent or branched origins of distant metastases and primary CRC[5,21], as well as those of other tumor types[22]. Consistent with these results, our data support that CRC BM tissues exhibit a diverse mutational pattern, likely evolved independently of their respective primary tissues. These extensive genomic differences emphasized the importance of metastatic biopsies during treatment. Therapy for CRC BM patients should be decided according to the genomic characteristics of metastatic biopsies rather than those of the primary tumor. In practice, the metastatic biopsies were difficult to obtain. Thus, circulating tumor cells may have a huge potential in the early detection and treatment of brain metastasis[23] and warrant further investigation.

Additionally, our results revealed an important role of DDR deficiency in the development of CRC BM. DDR deficiency in BM could be an early event and evolves independently from the primary CRC, which accumulates numerous yet distinct aging-related mutations. Theoretically, BM could be detected early by a characteristic DDR deficiency signal if the seeds are captured effectively in the blood. Thus, a well-designed prospective study on the application of circulating tumor cells for BM detection is warranted to better evaluate the dynamic evolution of BM of CRC.

Importantly, our study suggests evaluating biomarker-based therapy in CRC BM patients. Because numerous DDR drugs have been approved or evaluated recently, BM-selective HRD might provide a therapeutic target. HRD is most commonly found in breast cancers with genetic alterations of *BRCA1* and *BRCA2*. We identified functional *BRCA1/BRCA2* mutations selectively in CRC BM. The HRD mutational signature has recently been proposed as an ideal biomarker for NHEJ inhibitors[24], such as PARP inhibitors, which might be potentially useful in HRD BM patients.

In this study, we observed an enrichment of Signature 17 in metastasis tissues. Signature 17 was once identified in breast and metastatic cancer, but the etiology remains unknown. It was reported as an MMRD-related signature very recently, and we observed a significant correlation between this signature and the mutation burden in metastatic patients. The MMRD mutational signature has recently been regarded as an indicator for anti-PD-1

therapy[25], requiring a high degree of tumor infiltration by T lymphocytes ($CD4^+$ and/or $CD8^+$). MMRD is rarely found in primary brain tumors, and the microenvironments can vary greatly across tissues[26]. Little is known if MMRD leads to increased infiltration of lymphocytes in the brain. We identified elevated MMRD signals and T-cell infiltration in one BM tissue, suggesting immunotherapy as a potential choice for this subgroup of BM patients. However, no BM-specific mutation in mismatch repair genes was identified. There are several likely explanations: (1) the signature may have been caused by mutations in other DDR genes; (2) the FFPE tissues limits the power to detect mutations in all subclones; or (3) altered DDR gene or protein expression is caused by promoter methylation or posttranslational mechanism. Nevertheless, the mutational signature appears to serve as a sensitive and practical biomarker in FFPE tissues.

Chemo- or radiotherapy may also influence the genome instability level. In our study, all patients received chemotherapy, and 9 received radiation prior to the brain metastasis. Thus, we involved patients with liver metastasis, who also received chemo- and/or radiotherapy before metastasis. No significant difference was found between the HRD and MMRD signature levels of liver metastasis and primary tissues, indicating that the treatment before metastasis may not lead to the great difference observed in the study and that the elevation of HRD and MMRD levels may be brain metastasis specific. However, the evidence of liver metastasis is indirect and further studies involving brain metastatic CRC samples before treatment are warranted for the illumination of the role of HRD- and MMRD-related mutations in the BMs, though the collection of samples are difficult.

Finally, our study identified several potential drivers of CRC BM, such as voltage-gated sodium channels (*SCN7A* and *SCN2A*), which are important in neuronal excitability and are linked to metastasis and poor survival in multiple cancers[27]. Specially, recurrent founding mutations in *SCN7A* indicated that it could be important in the metastasis. *PDZRN4* has been identified as a tumor suppressor in human liver cancer[28], and recurrent mutations in the COSMIC database support its functional significance. The significance of these alterations clearly needs to be established in large cohorts and by functional studies. In addition, we observed that the majority of brain metastases studied (14/19) are from rectal primary cancers, which were consistent with previous findings[29,30]. Thus, genes with relatively higher mutation rate in our BM patients than in TCGA samples may provide additional clues for the underlying mechanisms of more brain metastasis in the primary rectal cancers.

In summary, we described mutational signatures and the tumor BM evolution process in 23 CRC patients. We identified characteristic mutational signatures of DDR deficiency (HRD and MMRD) in BM and suggested that the unstable features emerged in the early stage of CRC. Our findings reveal the vast genomic difference between metastatic and primary tissues, and suggest that PARP inhibitors and anti-PD-1 drugs may have clinical potential to prevent and treat DDR-deficient BM. We also identify potential BM drivers that might provide candidate biomarkers or targets for molecular therapies.

## Methods

**Patients and specimens**. The study protocol was reviewed and approved by the Ethics Committee of Jiangsu Province Hospital, the First Affiliated Hospital of Nanjing Medical University. We collected 54 tissue samples from 15 patients with brain metastatic CRC at Jiangsu Province's Hospital, 9 tissue samples from 3 patients with brain metastatic CRC at The First Affiliated Hospital of Zhejiang University, 3 tissue samples from 1 patient with brain metastatic CRC at Zhejiang Cancer Hospital, and 15 tissue samples from 5 patients with liver metastatic CRC at Jiangsu Province's Hospital. The trios of samples were taken from primary colorectal cancer, adjacent normal tissue, and brain metastases. For 9 patients, two primary tissues were sequenced. HE-stained sections from each sample were

subjected to an independent pathology review to confirm that the tumor specimen was histologically consistent with metastatic CRC (>40% tumor cells) and that the adjacent tissue specimen contained no tumor cells. The clinical information of all 19 patients is listed in Supplementary Table 1. RNA-Seq was conducted on the primary and BM tissues from two additional patients from Jiangsu Province's hospital and the clinical information is listed in Supplementary Table 1. We included 13 WES data sets in the same sample trio format from 4 CRC patients with brain metastases from a previous study[6]. For microsatellite instability (MSI) analysis, we included 41 trios of WES data from 5 patients with liver metastases[31]. We also collected 15 trios of tissue samples from 5 patients with liver metastatic CRC at Jiangsu Province Hospital (Supplementary Table 1). All participants provided written informed consent for genetic analysis.

**Next-generation sequencing.** For each individual, the genomic DNA of cells from one or two regions of the primary tumor, one brain or liver metastasis and one matched normal tissue sample was sequenced. DNA was extracted from formalin-fixed and paraffin-embedded (FFPE) tissue blocks using QIAGEN QIAAmp DNA extraction kits.

FFPE RNA was extracted using the Wuxi in-house method after FFPE sample sections were scalped into 1.5 mL microcentrifuge tubes, and then deparaffinization solution was used to remove paraffin. The RNA integrity was determined using a 2100 Bioanalyser (Agilent) with $DV_{200}$ (Percentage of RNA fragments > 200 nt fragment distribution value) and quantified by the NanoDrop (Thermo Scientific).

WES: Library construction and whole-exome capture of genomic DNA were performed using the SureSelectXT reagent kit (Agilent) and SureSelectXT Human All Exon V5 (Agilent) on an Agilent Bravo Automated Liquid Handling Platform (Agilent). The captured DNA was sequenced on an Illumina HiSeq 2500 sequencing platform, with 125-bp paired-end sequencing.

WGS: We sequenced 24 tissues on the Illumina HiSeq X Ten instrument run in the standard mode in WuXi NextCODE (Shanghai) Co., Ltd, with 150-bp paired-end sequencing. SeqPlus service (https://www.wuxinextcode.com/product/seqplus/) was used to generate high-quality data with a low level of artifacts.

RNA-Seq: RNA purification, reverse transcription, library construction and sequencing were performed at WuXi NextCODE at Shanghai according to the manufacturer's instructions (Illumina). The captured coding regions of the transcriptome from total RNA were prepared using the TruSeq® RNA Exome Library Preparation Kit. For FFPE samples, the RNA input for library construction was determined by the quality of RNA. Generally, 20 ng of RNA was recommended for high-quality RNA, and 20–40 ng of RNA was recommended for medium-quality FFPE RNA. For low-quality RNA, 40–100 ng of total RNA was used as the input. Next, the cDNA was generated from the input RNA fragments using random priming during first- and second- strand synthesis, and sequencing adapters were ligated to the resulting double-stranded cDNA fragments. The coding regions of the transcriptome were then captured from this library using sequence-specific probes to create the final library. After the library was constructed, the Qubit 2.0 fluorometer dsDNA HS Assay system (Thermo Fisher Scientific) was used to quantify the concentration of the resulting sequencing libraries, while the size distribution was analyzed using the Agilent BioAnalyzer 2100 system (Agilent). Sequencing was performed using an Illumina system following Illumina-provided protocols for 2 × 150 paired-end sequencing in WuXi NextCODE at Shanghai, China.

**Sequencing alignment and detection of somatic variants.** The FastQC package (http://www.bioinformatics.babraham.ac.uk/projects/fastqc) was used to assess the quality-score distribution of the sequencing reads. Read sequences were mapped to the human reference genome (GRCh37) using Burrows-Wheeler Aligner (BWA-MEM v0.7.15-r1140)[32] with the default parameters, and duplicates were marked and discarded using Picard (v1.70) (http://broadinstitute.github.io/picard). After alignment by BWA, the reads were subjected to local realignment and recalibration using the Genome Analysis Toolkit (GATK v3.5). For the 4 patients from a previous study, we downloaded raw FASTQ files from dbGaP (https://www.ncbi.nlm.nih.gov/projects/gap/cgi-bin/study.cgi?study_id = phs000730.v1.p1) and applied the sample alignment strategy accordingly.

Somatic substitutions and indels were detected using the MuTect2 mode in GATK on the GRCh37 genome build following the best practices for somatic SNV/indel calling (https://software.broadinstitute.org/gatk/best-practices/). Briefly, the algorithms compared the tumor with the matched normal sample to exclude germline variants. Somatic calls were excluded (1) if they were found in a panel of normal controls assembled from matched normal tissues, (2) if they were located in the segmental duplication region marked by the UCSC browser (http://genome.ucsc.edu/), or (3) if they were found in the 1000 Genomes Project (the Phase III integrated variant set release, across 2,504 samples) with the same mutation direction. Mutations (SNVs/indels) were annotated with the local versions of Oncotator (1.8.0)[33] according to GENCODE v19. We also applied the ffpe-filter module of ngs-filter (https://github.com/mskcc/ngs-filters) to remove C > T mutations after somatic mutation calling. In addition, the annotated somatic mutations located in exon of SCN7A were further confirmed by sanger sequencing. The primer sequence is as follows: forward primer 5′-CTCTTTGCTGCCTCCATCAC-3′ and reverse primer 5′-CTGAATTGAGCAACACCCAAGA-3′.

For comparison, we downloaded the mutation data (maf files) of 489 colorectal cancer patients from the Firehose Broad GDAC (http://gdac.broadinstitute.org/, version 2016_01_28).

RNA reads were generated, aligned to the GENCODE v19 genome assembly with STAR v2.4.1[34] and quantified using featureCounts v1.5.0[35]. The raw read counts were normalized by DESeq2[36] followed by differential expression analysis. Gene set enrichment analysis (GSEA) was performed using the log fold change between brain metastasis and primary tissues using the GSEA command from the Bioconductor package clusterProfiler[37] based on DNA damage response genes curated from a previous review[14].

**Mutational signature analysis.** We converted all mutation data from WES and WGS data sets into a matrix (M) composed of 96 features comprising mutation counts for each mutation type (C > A, C > G, C > T, T > A, T > C and T > G) using each possible 5′ and 3′ context for all samples and applied the R package deconstructSigs[38] to infer the proportion of each known mutational signature. In total, 30 signatures reported by COSMIC (http://cancer.sanger.ac.uk/cosmic/signatures) were included in the analysis (Supplementary Data 1). The mutation number and the mutation burden were provided in Supplementary Table 2. Mutation burden was defined as the number of somatic mutations per megabase (/MB) of genome examined.

**Estimation of the copy number and cancer cell fraction.** The GATK best practices for somatic copy number alterations (CNAs) in exomes were used to detect CNAs from the whole-exome sequencing data (https://software.broadinstitute.org/gatk/best-practices/). The somatic copy number was estimated by ReCapSeg, which is implemented as part of GATK (v4). Briefly, the read counts for each of the exome targets were divided by the total number of reads to generate proportional coverage. A panel of normal (PON) controls was built using proportional coverage from 19 normal samples. Each of the tumor samples was compared with the PON, after which tangent normalization was applied. Circular binary segmentation (CBS) was then applied to segment the normalized coverage profiles. Sex chromosomes (X and Y) were excluded from this analysis. The cancer cell fraction for each mutation was estimated using ABSOLUTE v1.2 based on the variant allele frequency (VAF) and segmented copy number data[6]. The purity of each sample is provided in the Supplementary Table 2.

**Evolution and subclonality analysis.** We included 353 human DDR genes curated by experts in a previous review[14]. All mutations in DDR genes were classified into three categories according to their cancer cell fraction (CCF) in brain metastases and primary specimens: (1) mutations only in BMs, defined by $CCF_{BM} > 0$ and $CCF_{primary} = 0$; (2) mutations only in primary tumors, defined by $CCF_{BM} = 0$ and $CCF_{primary} > 0$; and (3) shared mutations, defined as other mutations with $CCF_{BM} > 0$, $CCF_{primary} > 0$.

In 11 trios with WES data, we also conducted subclonality analysis and inferred clonal evolution. The number of subclones contributing to a sample and relative contribution was estimated using the Bioconductor package sciClone[39] based on the variant allele frequency (VAF). CCF values were converted to the sciClone input (divided by 2 such that clonal mutations have a value of 0.5)[40]. A minimum tumor sequencing depth of 50 was required for each mutation. Next, ClonEvol packages were used for phylogenetic inference from CCF subclones and following visualization[41]. Briefly, this tool first enumerates all trees independently for each sample and then tries to build a 'consensus' tree model that fits multiple samples from a single patient at once. We successfully obtained consensus models in 7 of 11 trios and constructed phylogenetic trees accordingly.

**Detection of microsatellite instability.** To further validate the signature of MMRD, we ran MSIsensor[42] to infer the somatic MSI level by comparing microsatellite distributions in tumors and their matched normal samples. We applied the optimal MSIsensor parameters as below:[43] msisensor msi -d microsatellites.list -n normal.bam -t tumor.bam -q 2 -b 4 -f 0.1.

As suggested by the literature, patients with MSI scores greater than 3.0 were regarded as MSI; all others were considered microsatellite stable.

**Immunohistochemistry analysis.** Immunohistochemistry was used to estimate the abundance of CD4 + cells and CD8 + T cells in BM tissues. Following the process of dewaxing and rehydration, sections were placed in a retrieval solution (G1202, pH 6.0) and then heated to boiling for ~25 min. As the sections cooled to room temperature, they were submerged in phosphate-buffered saline solution (pH 7.4) three times. The slides were further processed on an automated immunostainer. Endogenous peroxidase was blocked with 3% $H_2O_2$ solution shielded from light at room temperature. Next, the slides were blocked with 3% BSA to avoid nonspecific background staining. Thereafter, the specimens were incubated overnight at 4 °C with primary antibodies (MAB 0039 and MAB 0021, for CD4 and CD8, respectively; MXB Biotechnology, Fuzhou, China) and then were incubated for 1 h at room temperature with biotinylated secondary antibodies diluted to 1:1000. The samples were developed using a DAB-Plus Substrate Kit (Invitrogen) under a microscope and were counterstained with hematoxylin, a nucleophilic dye. Finally, after dehydration and mounting, the specimens were

observed under a microscope and imaged for further analysis. To ensure the consistency of the results, we randomly selected 10 representative images for different levels of tumor-infiltrating lymphocytes (TILs), and the results were assessed by three pathologists blinded to the patients' clinical information.

**Statistical analysis and figures**. The median was used if multiple samples from the same tissues were sequenced. All statistical tests were performed using a Wilcoxon rank-sum test for continuous data. Fisher's exact test was used to assess differences in the count data. We applied the MatchIt R package for sample matching[44]. Multiple testing corrections were performed where necessary using the Benjamini–Hochberg method. All reported *P* values were two-sided. Mutational lolliplots were generated by ProteinPaint[45]. Other figures were generated using the R package ggplot2 and RColorBrewer.

**Reporting Summary**. Further information on research design is available in the Nature Research Reporting Summary linked to this article.

## Data availability

The whole-exome sequencing data and the whole-genome sequencing data have been deposited in the European Genome-phenome Archive (EGA) at the EMBL-European Bioinformatics Institute under accession number EGAD00001005030. The RNA-Seq data have been deposited at Gene Expression Omnibus (GEO) under accession numbers GSE132226. All the other data supporting the findings of this study are available within the article and its supplementary information files and from the corresponding author upon reasonable request.

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

## Acknowledgements

This work was supported by National Natural Science Foundation of China (No 81572389 and No 81871944 to Y.G, No 81302109 to J.S, No 81703295 to C.W and No 81472210 to W.F); Jiangsu 333 Project (BRA2016517 to Y.G); and Jiangsu Province Key Medical Talents (ZDRCA 2016026 to Y.G).

## Author contributions

J.S., C.W., Y.S., H.S., Z.H. and Y.G. conceived the original idea of the work, designed and performed the experiments, and conducted the data analysis and interpretation. J.S., Y.S.,

H.S. and Y.G. provided financial support and administrative support, too. Y. Zhang, X.H., W.H., P.L., W.F., Y. Zhu, Y. Zheng, J.Z., J.Y., Z.M., X. Cai, B.W. provided study materials or patients. X.X., L.X., X. Chen, Y. Zheng, X.H., W.H. participated in collection and assembly of data. All authors contributed to writing the manuscript and final approval of the manuscript.

## Additional information

**Competing interests:** The authors declare no competing interests.

