## [Peer Review File · Nature Communications]

Reviewers' comments:

Reviewer #1 (Remarks to the Author):

Brain metastases from colorectal cancer are rare. This group of investigators looked at 15 brain metastases, primary tumors and normal tissue from colorectal cancer and compared to TCGA. They report that BM have elevated signatures of HRD and MMRD compared to primary CRC. They also report that DDR signatures seem to be enriched in brain metastases and occur early, and mutations in DDR genes are specific to brain metastases in some cases.

- 1) This study would be significantly strengthened with larger sample numbers.
- 2) When reporting the absence of specific mutations in the primary tumor, can the authors provide power calculations for their power to be able to detect a mutation at that particular locus?
- 3) When comparing TCGA to this cohort of brain metastases, did the authors correct for differences in the cohorts (demographics, sequencing techniques, clinical variables such as age, stage at diagnosis, exposure to treatment)?
- 4) Were the differences in the frequency of mutations between brain metastases and the primary tumors in the DDR genes statistically significant?
- 5) All patients received chemotherapy and 6 received radiation prior to the brain metastasis. Can the differences observed be because of treatment?

Reviewer #2 (Remarks to the Author):

The authors of the manuscript "Genomic Signatures Reveal DNA Damage Response Deficiency in Brain Metastases of Colorectal Cancer" reveal the mutational signatures in a rare but lethal disease Brain Metastases of Colorectal Cancer by whole exome sequencing. Their research start from 11 CRC patients. Three tissues, including primary colorectal cancer, adjacent normal tissue and brain metastases, were sequenced. In their results, they identified some mutations, which been linked to aging, homologous recombination deficiency (HRD), and mismatch repair deficiency (MMRD), in either BM or primary tissues. They also investigated the evolution of somatic mutations in DNA damage response genes. Two BM-only heterozygous BRCA2 mutations were identified in one patient. In general, this manuscript aims to resolve a common question in cancer whether mutations impact tumors and their metastases. They reveal the importance of DNA damage response genes in the brain metastasis of CRC. However, this paper may be limited to its sample size that is key to come to a convincing conclusion. The authors provide the observations of BM-related genes, but they did not provide evidences that the mutations in these related genes could cause BM. In my opinion, this work may provide some new findings, but it can be improved if they can obtain more samples and validate their results for the journal. I have some questions in my mind after reading this manuscript:

- 1) The authors analyzed 6 mutations groups, do they checked other mutations?
- 2) I am very interested in the gene or protein expression of the BM related genes. For example, DDR genes, do they still express in brain metastases of CRC?
- 3) Are any CRC BM related mutations or genes identified before? If some are known, it can be mentioned in introduction.

Reviewer #3 (Remarks to the Author):

Metastasis to the brain from a colorectal cancer primary tumor is uncommon but associated with a

particularly poor prognosis. The molecular understanding of this metastatic tumor type is also poor and is required to improve therapeutic approaches. This manuscript describes the results of sequencing the exomes of matched normal, primary and metastatic tissue from 15 patients with metastatic brain cancer from a colorectal primary. The collection of such rare and difficult to attain samples increases the novelty of this contribution. The clinical challenges of collecting such tissue also mean that all samples were archival (FFPE) and metastatic samples were post therapy, which does increase experimental difficulty and interpretation. The major finding of an increased rate of DNA damage response deficiency may have clinical significance in terms of choice of therapy as well as choice of biopsy site for informing therapeutic decisions.

Based on my understanding, the analysis and statistical methods employed seem largely appropriate and standard to these types of analyses. Additional review by an expert bioinformatician would be desirable. Of particular importance are to seek additional opinion on the genomic signatures analyses. Genomic signatures are generally reported for individual tumors, rather than averaged for a series of tumors (as is presented in Figure 1). The way the data in Figure 1 is currently presented makes it difficult to interpret the contribution of each signature to individual tumors. The authors should also expand the reported details for this figure, eg. the number of mutations identified for each individual tumor. Based on this, was there adequate power to assess mutational signatures? This is not always possible from exome (rather than whole genome) data. The over-representation of C>T mutations may be due to the use of FFPE samples. This can be improved by use of an additional DNA treatment step, although it does not seem from the methods that this was conducted.

The tumour cellularity (overall, for each sample) is not reported, which is important for interpretation of the data. It is unusual to see the 5 category classification for DDR genes based on cancer cell fraction, without correcting for cellularity. Categories 1, 3 and 5 seem reasonable, however the 'mainly' categories (2 and 4) are unusual and the reason for the 0.5 cut-off is not clear. (lines 184-192).

It is a strength of the study that multiple biopsies have been used for individual patients, however it seems a missed opportunity to explore clonal evolution. How similar were biopsies from the same tumors taken at the same time? It does not seem appropriate to average these.

All brain metastases were taken after chemotherapy. This should be highlighted in the main body of the paper (not just the supp clinical info). Were there any patterns based on type of therapy or radiation? Is it thought that the differences between the primary tumors and brain metastases arose due to chemotherapy or disease progression?

The extensive genomic differences identified between primary and metastatic tumors is very important and should be highlighted more strongly in terms of clinical implications, eg. that it may be more appropriate to inform therapies based on circulating cells or metastatic biopsies rather than the primary tumor.

How did the MMR signature compare to mutational load? Were these concordant?

Minor comments:

Number of biopsies – line 101 states 59 biopsies, however in methods it seems there are 42 biopsies from 11 patients and a further 13 from 4 patients, giving a total of 59, could you please clarify?

There are some minor grammatical errors throughout

Abbreviations are sometimes used without clearly defining, eg CCF

Figure legends require further details to explain what is being presenting, including to explain the meaning of different colours used

Reviewers' comments:

Reviewer #1 (Remarks to the Author):

Brain metastases from colorectal cancer are rare. This group of investigators looked at 15 brain metastases, primary tumors and normal tissue from colorectal cancer and compared to TCGA. They report that BM have elevated signatures of HRD and MMRD compared to primary CRC. They also report that DDR signatures seem to be enriched in brain metastases and occur early, and mutations in DDR genes are specific to brain metastases in some cases.

1) This study would be significantly strengthened with larger sample numbers.

Response: In response to the reviewer's comment, we increased the sample size in the revised version. We conducted whole-genome sequencing on "trios" (brain metastasis, colorectal primary and adjacent normal) tissues from 8 independent patients (Supplementary Table 1). We observed the consistent elevation of HRD and MMRD signatures in the brain metastasis of these patients (Figure 1). We also find an additional Signature (COSMIC signature 17) was also elevated in brain metastases. The signature was recently proved to emerge after the deficiency of MMR¹, further support our findings. In addition, we performed whole-exome sequencing on "trios" (liver metastasis, colorectal primary and adjacent normal) tissues from 5 additional patients (Supplementary Table 2), indicating that the level of HRD and MMRD in liver metastasis did not increase significantly (Supplementary Figure 3 and Figure 3). Thus, the increase of DNA damage may not be derived from therapy.

2) When reporting the absence of specific mutations in the primary tumor, can the authors provide power calculations for their power to be able to detect a mutation at that particular locus?

Response: In response to the reviewer's comment, we provide the power calculated by ABSOLUTE on the reported mutations in the Supplementary Table 5.

3) When comparing TCGA to this cohort of brain metastases, did the authors correct for differences in the cohorts (demographics, sequencing techniques, clinical variables such as age, stage at diagnosis, exposure to treatment)?

Response: No, we did not correct for differences in the cohorts in the last version. In the revision, we match the age, gender and stage (1:8) to correct for the difference using nearest matching from MatchIt R package². There is no great difference and we provided the results in the Supplementary Figure 5. We did not correct for the race because most of the TCGA samples were whites or black Africans. Both our data and TCGA data involved whole-exome sequencing, thus we did not correct for sequencing techniques. Because most TCGA tissues were collected before treatment, we did not correct for treatment therapy.

4) Were the differences in the frequency of mutations between brain metastases and the primary tumors in the DDR genes statistically significant?

Response: The frequency of DDR mutations was slightly higher in brain metastases than in the primary tumors but the difference was not significant (Response Figure 1). The low mutation frequency and heterogeneous function of DDR genes may be responsible for the insignificant

results.

Response Figure 1. The difference of DDR mutation frequency between brain metastases and primary tumors.

5) All patients received chemotherapy and 6 received radiation prior to the brain metastasis. Can the differences observed be because of treatment?

Response: We thank the reviewer for raising this important point. Chemo- or radiotherapy may influence the genomic characters of tissues. But it is very difficult to collect tissues from patients without any chemo- or radiotherapy before the metastasis. Thus, we prove that the treatment before metastasis may not lead to the great difference observed in the study by investigating the difference of genomic features in patients with liver metastasis, who also received chemotherapy and/or radiotherapy before metastasis. In the revised version, we included tissues from a total of 10 patients with liver metastasis (liver metastasis, colorectal primary and adjacent normal) tissues. We found that the level of HRD and MMRD signature did not increase significantly in liver metastasis. We discussed the results in Page 21-22 (lines 447-454).

Reviewer #2 (Remarks to the Author):

The authors of the manuscript “Genomic Signatures Reveal DNA Damage Response Deficiency in Brain Metastases of Colorectal Cancer” reveal the mutational signatures in a rare but lethal disease Brain Metastases of Colorectal Cancer by whole exome sequencing. Their research start from 11 CRC patients. Three tissues, including primary colorectal cancer, adjacent normal tissue and brain metastases, were sequenced. In their results, they identified some mutations, which been linked to aging, homologous recombination deficiency (HRD), and mismatch repair deficiency (MMRD), in either BM or primary tissues. They also investigated the evolution of somatic mutations in DNA damage response genes. Two BM-only heterozygous BRCA2 mutations were identified in one patient. In general, this manuscript aims to resolve a common question in cancer whether mutations impact tumors and their metastases. They reveal the importance of DNA damage

response genes in the brain metastasis of CRC. However, this paper may be limited to its sample size that is key to come to a convincing conclusion. The authors provide the observations of BM-related genes, but they did not provide evidences that the mutations in these related genes could cause BM. In my opinion, this work may provide some new findings, but it can be improved if they can obtain more samples and validate their results for the journal.

Response: We thank the reviewer for the encouraging comments and suggestions that help enhance the value of our study. In response to the reviewer's comment, we increased the sample size in the revised version. We conducted whole-genome sequencing on "trios" (brain metastasis, colorectal primary and adjacent normal) tissues from 8 independent patients (Supplementary Table 1). We observed the consistent elevation of HRD and MMRD signatures in the brain metastasis of these patients (Figure 1). In addition, we performed whole-exome sequencing on "trios" (liver metastasis, colorectal primary and adjacent normal) tissues from 5 additional patients (Supplementary Table 2), further supporting that the level of HRD and MMRD in liver metastasis did not increase significantly (Supplementary Figure 3 and Figure 3).

I have some questions in my mind after reading this manuscript:

1) The authors analyzed 6 mutations groups, do they checked other mutations?

Response: We thank the reviewer for the helpful question. In the last version, we checked all mutational signatures but presented the first signatures for visualization. In the revised version, we also investigated all 30 cosmic mutational signatures in the additional whole-genome sequencing data. Interestingly, we found an additional signature (Cosmic Signature 17) elevated in brain metastasis. The signature 17 was commonly observed in the breast cancer and metastasis tissues and it can occur in a model organism with MMRD. In our data, the proportion of the signatures was significantly correlated with mutation burden ($r=0.58$, $P=1.72 \times 10^{-5}$, Supplementary Figure 2). We also presented the average proportion of all other signatures in Supplementary Table 3.

2) I am very interested in the gene or protein expression of the BM related genes. For example, DDR genes, do they still express in brain metastases of CRC?

Response: We agree with the reviewer that the gene or protein expression of the brain metastasis-related genes are important. However, the collection of proper brain metastasis tissues is difficult and time-consuming. It took us more than a year to obtain the tissues from 8 brain metastasis patients used for whole-genome sequencing in this revision. Unfortunately, we did not have additional samples for the detection of RNA or protein expression. But the expression of DDR genes in brain metastasis warrants further investigation and we will explore in depth the issue in the future study.

3) Are any CRC BM related mutations or genes identified before? If some are known, it can be mentioned in introduction.

Response: We thank the reviewer for the suggestion. Several clinical investigations have reported higher mutation rate of *RAS* genes and *PIK3CA* in brain metastasis tissue^{3,4}. We have introduced the results in the Introduction (Page 6, lines 112-114).

Reviewer #3 (Remarks to the Author):

Metastasis to the brain from a colorectal cancer primary tumor is uncommon but associated with a particularly poor prognosis. The molecular understanding of this metastatic tumor type is also poor and is required to improve therapeutic approaches. This manuscript describes the results of sequencing the exomes of matched normal, primary and metastatic tissue from 15 patients with metastatic brain cancer from a colorectal primary. The collection of such rare and difficult to attain samples increases the novelty of this contribution. The clinical challenges of collecting such tissue also mean that all samples were archival (FFPE) and metastatic samples were post therapy, which does increase experimental difficulty and interpretation. The major finding of an increased rate of DNA damage response deficiency may have clinical significance in terms of choice of therapy as well as choice of biopsy site for informing therapeutic decisions.

Based on my understanding, the analysis and statistical methods employed seem largely appropriate and standard to these types of analyses. Additional review by an expert bioinformatician would be desirable. Of particular importance are to seek additional opinion on the genomic signatures analyses. Genomic signatures are generally reported for individual tumors, rather than averaged for a series of tumors (as is presented in Figure 1). The way the data in Figure 1 is currently presented makes it difficult to interpret the contribution of each signature to individual tumors. The authors should also expand the reported details for this figure, eg. the number of mutations identified for each individual tumor. Based on this, was there adequate power to assess mutational signatures? This is not always possible from exome (rather than whole genome) data.

Response: We thank the reviewer for the suggestion. We have added a supplementary table (Supplementary Table 3) to present mutation number, the proportion of mutational signatures and purity of each sample in this revision. We identified an average of 300 mutations for whole-exome sequencing data. In this revision, we also conducted whole-genome sequencing on “trios” (brain metastasis, colorectal primary and adjacent normal) tissues from 8 independent patients (Supplementary Table 1) and observed consistent results.

The over-representation of C>T mutations may be due to the use of FFPE samples. This can be improved by use of an additional DNA treatment step, although it does not seem from the methods that this was conducted.

Response: In this study, we used SeqPlus service (<https://www.wuxinextcode.com/product/seqplus/>) to generate the high-quality data with a low level of artifacts. We also applied ffpe-filter module of ngs-filter (<https://github.com/mskcc/ngs-filters>) to remove C>T mutations after the somatic mutation calling. We have mentioned them in the Methods (Page 8-9). In addition, we included FFPE samples for both primary and metastatic tissues. The preparation of FFPE samples followed the sample protocol. A paired difference test was used to evaluate the difference between primary and metastatic tissues. Thus, we thought that the over-presentation of C>T mutations may not be due to the use of FFPE samples.

The tumour cellularity (overall, for each sample) is not reported, which is important for interpretation of the data. It is unusual to see the 5 category classification for DDR genes based on

cancer cell fraction, without correcting for cellularity. Categories 1, 3 and 5 seem reasonable, however the ‘mainly’ categories (2 and 4) are unusual and the reason for the 0.5 cut-off is not clear. (lines 184-192).

Response: We thank the reviewer for the helpful suggestion. In the revision, we have provided tumor purity estimated by ABSOLUTE in the Supplementary Table 3. Sample purity and local copy-number were used to correct mutant and reference read fractions to estimate cancer cell fraction⁵. For the category of DDR genes in Figure 2, we agreed with the reviewer that the cut-off of 0.5 is arbitrary and only a few mutations were classified in categories 2 and 4. Thus, we removed the categories 2 and 4 and regenerated the figure in this revision (Figure 2).

It is a strength of the study that multiple biopsies have been used for individual patients, however it seems a missed opportunity to explore clonal evolution. How similar were biopsies from the same tumors taken at the same time? It does not seem appropriate to average these.

Response: We understand the reviewer's concerns. Clonal evolution analysis prefers an ultra-deep depth sequencing (>500X) and multiple biopsies in a single patient to determine a proper model from a number of potential evolutionary models. In this study, we conducted general depth sequencing (60-100X) on at most two biopsies for a patient, thus we did not include clonal evolution inference for each individual but mainly focused on the change of genomic signatures in brain metastasis.

For the analysis of mutational signatures, we calculated the Spearman correlation between the proportions of 30 mutational signatures in patients with two biopsies and we found that most patients (7/9, 78%) carried similar mutational signatures ($\rho > 0.5$) in different biopsies. Thus, we average the results for further analysis. For the analysis of specific mutations, we examined the cancer cell fraction of each biopsy separately.

All brain metastases were taken after chemotherapy. This should be highlighted in the main body of the paper (not just the supp clinical info). Were there any patterns based on type of therapy or radiation? Is it thought that the differences between the primary tumors and brain metastases arose due to chemotherapy or disease progression?

Response: We thank the reviewer for raising this important point. Chemo- or radiotherapy may influence the genomic characters of tissues. But it is very difficult to collect tissues from patients without any chemo- or radiotherapy before the metastasis. Thus, we prove that the treatment before metastasis may not lead to the great difference observed in the study by investigating the difference of genomic features in patients with liver metastasis, who also received chemotherapy and/or radiotherapy before metastasis. In the revised version, we included tissues from a total of 10 patients with liver metastasis (liver metastasis, colorectal primary and adjacent normal) tissues. We found that the level of HRD and MMRD signature did not increase significantly in liver metastasis. We discussed the results in Page 21-22 (lines 447-454).

The extensive genomic differences identified between primary and metastatic tumors is very important and should be highlighted more strongly in terms of clinical implications, eg. that it may be more appropriate to inform therapies based on circulating cells or metastatic biopsies rather than the primary tumor.

Response: We thank the reviewer for valuable suggestion. We have emphasized the importance of

the difference in the discussion (Page 20, lines 407-414):

“Consistent with these results, our data support that CRC BM tissues exhibit a diverse mutational pattern, likely evolve independently of their respective primary tissues. These extensive genomic difference emphasized the importance of metastatic biopsies during the treatment. The therapy for CRC BM patients should be decided according to the genomic characters of metastatic biopsies rather than the primary tumor. In practical, the metastatic biopsies were difficult to obtain. Thus, circulating tumor cells may have huge potential in the early detection and treatment of brain metastasis⁶ and warranted further investigation.”

How did the MMR signature compare to mutational load? Were these concordant?

Response: We thank the reviewer for the insightful suggestion. In the revised version, we determined a new signature (COSMIC signature 17), which was recently proved to emerge after the deficiency of MMR¹. Interestingly, we found that the proportion of signature 17 was significantly correlated with the mutation rate ($r=0.58$, $P=1.72\times 10^{-5}$, Supplementary Figure 2)

Minor comments:

Number of biopsies – line 101 states 59 biopsies, however in methods it seems there are 42 biopsies from 11 patients and a further 13 from 4 patients, giving a total of 59, could you please clarify?

Response: Sorry for unclear description. We have clarified the description in Page 6-7 (lines 115-120) as suggested by the reviewer.

“In this study, we conducted whole-exome sequencing and whole-genome sequencing (WGS) or of biopsy samples from “trios” of patient-matched brain metastases, primary CRC tissues and adjacent normal samples (WES: 42 tissues from 11 patients; public WES: 13 tissues from 4 patients; WGS: 24 tissues from 8 patients) to (1) document genomic signatures during the evolution of CRC BM; (2) identify clinically actionable targets for the BM treatment; and (3) identify potential novel drivers in CRC BM.”

There are some minor grammatical errors throughout

Response: Sorry for the errors. In this revision, we have carefully reviewed the manuscript and proofed the grammatical errors.

Abbreviations are sometimes used without clearly defining, eg CCF

Response: Sorry for unclear definition. In the revised version, we define the cancer cell fraction (CCF) at first mention (Page 11). We also examined all abbreviations carefully.

Figure legends require further details to explain what is being presenting, including to explain the meaning of different colours used

Response: Sorry for unclear figure legends. In the revised version, we added the necessary details in the figure legends.

Reference

1. Meier B, *et al.* Mutational signatures of DNA mismatch repair deficiency in *C. elegans* and human cancers. *Genome research* **28**, 666-675 (2018).
2. Ho D, Imai K, King G, Stuart EA. MatchIt: Nonparametric Preprocessing for Parametric Causal Inference. *2011* **42**, 28 (2011).
3. Yaeger R, *et al.* RAS mutations affect pattern of metastatic spread and increase propensity for brain metastasis in colorectal cancer. *Cancer* **121**, 1195-1203 (2015).
4. Tie J, *et al.* KRAS mutation is associated with lung metastasis in patients with curatively resected colorectal cancer. *Clinical cancer research : an official journal of the American Association for Cancer Research* **17**, 1122-1130 (2011).
5. Brastianos PK, *et al.* Genomic Characterization of Brain Metastases Reveals Branched Evolution and Potential Therapeutic Targets. *Cancer discovery* **5**, 1164-1177 (2015).
6. Massague J, Obenauf AC. Metastatic colonization by circulating tumour cells. *Nature* **529**, 298-306 (2016).

Reviewers' comments:

Reviewer #1 (Remarks to the Author):

In this paper, the authors analyze 19 trios of BM, CRC and normal tissue to identify potential BM related genes.

1. The authors should explore clonal evolution from these matched tumors in this sample cohort.
2. This paper would be strengthened with RNASeq and protein expression data as orthogonal tests to validate findings.
3. When the authors report that mutations were in the brain met only, and not in the primary tumor, how powered were they to detect that mutation in the primary? The authors have included power calculations but do not include this information. For example, the authors report that patients BM013 and BM014 carried BM-only mutations on RAD51B and PAXIP1. They need to give us power calculations for what the power was to detect those mutations in the primary tumor, in order to confirm they were absent. Similarly, they report that BM004 carried BM-only mutations in POLB gene How powered were they to detect mutations in POLB in the primary tumor These are not included in their power calculations in Supplementary Table 5. This is a critical part of this paper. When claiming BM only mutations, or primary tumor only mutations, we need to know how powered they were to detect these mutations in the other site where they were reported absent.
4. A number of grammatical and linguistic errors need to be corrected.

Reviewer #2 (Remarks to the Author):

I am glad to know the authors conducted whole-genome sequencing on "trios" tissues from 8 patients and provided some interesting results on signature analysis. Now this work includes 19 patients with brain metastatic CRC, 5 patients with liver metastatic CRC. Even though the manuscript has been much improved, I still have some concerns:

1. Chemo- and radiotherapy have been performed for the patients prior to brain metastatic in this study. The authors involved patients with liver metastasis to show that treatment didn't lead to great difference. I know it is difficult to obtain brain metastatic CRC samples before these treatment, but using liver metastasis cannot prove small difference in brain metastasis. Do the authors correct the mutations based on adjacent normal samples when you compare brain metastatic CRC samples with primary CRC samples? Normal cells may proliferate slower than tumor cells, so it could be less mutations in those.
2. In the reference 5 cited, Naxerova et al. reconstructed the evolutionary relationship of primary tumors, lymph node metastases, and distant metastases. They found that in 65% of cases, lymphatic and distant metastases arose from independent subclones in the primary tumor, whereas in 35% of cases they shared common subclonal origin. Do you conduct similar analysis?
3. The mutation patterns of different mutational signatures should be shown somewhere.
4. In figure 1A, C->T mutation in primary tumor is significantly higher than in brain metastatic. The authors cited reference 18, but I cannot find any description about C>T substitutions have been associated with epithelial tumors. In addition, C>T substitutions are very common in different mutational signatures. It looks it is not consistent with elevated mutational signatures in BM samples.
5. It is not clear what are aging-related mutations on line 288. It may be a typo for "the primary metastatic CRC samples".
6. On line 323, the authors found that TP53 was the genes with the highest mutation rate, but almost all TP53 mutations were shared between BM and primary tissues, suggesting that TP53 mutations are drivers and occur prior to BM. Do the authors check TP53 mutations in normal tissue samples?
7. The authors have shown comprehensive analysis in nucleotide level, also expression level for some of genes. They also show most of potential driver genes related to BM are missense mutations, for example, IKZF1. In figure 4B, it has been shown in TCGA data, the expression of

IKZF1 is significantly associated with poor survival of CRC patients. I would ask whether IKZF1 is expressed in normal tissues or other cancers. It would be interesting to see how IKZF1 is related to poor survival of CRC patients.

8. I am interested in how authors generated supplementary Figure 2. The brief method should be introduced in Methods or Figure legends, including other figures. What's the unit of signature 17 means in x axis?

9. On line 203, full name of CNA should be introduced.

Reviewer #3 (Remarks to the Author):

It is commendable that the authors have increased the sample number to compare liver metastasis samples. The manuscript could still be strengthened. It would be beneficial to graphically present mutational signatures for each tumor, not just the numerical values that are presented in the new supplemental data. Figure 1 image and legend are still not clear. More clinical data should be included in the methods section. It is especially necessary to know which therapies have been given prior to surgery. One of the main conclusions is that DNA damage signatures increased in metastatic brain cancer, however this may simply be due to therapy exposure. Supplementary table 1 could be substantially improved by including the type of chemotherapy given as well as the additional liver metastasis samples included in the revised version. It has also come to my attention that the majority of brain metastases studied (14/19) are from rectal primary cancers. Radiotherapy is usually given in the setting of rectal cancer. Only a single case is from the distal colon, which is the most common site of primary colorectal cancer. My concern is that this series is not representative of most colorectal cancers. Are rectal cancers known to more commonly metastasise to the brain? This may be reasonable, however should be highlighted in the manuscript. There are some differences in therapy regimen between the patients, did this correlate with mutational signatures?

Reviewers' comments:

Reviewer #1 (Remarks to the Author):

In this paper, the authors analyze 19 trios of BM, CRC and normal tissue to identify potential BM related genes.

1. The authors should explore clonal evolution from these matched tumors in this sample cohort.

Response: We thank the reviewer for the suggestion. In this revision, we inferred clonal evolution in 11 “trios” that had WES data with relatively deep depth and involved multiple samples from each primary tissue. For each “trio”, we first estimated the clonal number based on the cancer cell fraction and constructed the phylogenetic tree based on a consensus model obtained from primary and brain metastasis (BM) samples. In this analysis, we successfully obtained consensus models in 7 of 11 “trios” and constructed phylogenetic trees accordingly. Interestingly, we identified four BM founding mutations in DNA damage response genes (i.e., *BRCA2*, *POLB*, *CHEK1*, and *SLX4*). We also noticed that the two recurrent mutations in *SCN7A* (p.L805P/R) were also founding mutations of BM samples. These results indicated that these mutations should appear in the initial stage of brain metastasis and may contribute to the brain metastasis. We have added the methods (Page 13, lines 260-270) and the results (Page 18, lines 375-377 and lines 383-384; Page 19, lines 394-396; Pages 22, lines 459-461) in this revision.

2. This paper would be strengthened with RNASeq and protein expression data as orthogonal tests to validate findings.

Response: We thank the reviewer for the suggestion. Because it is very difficult to obtain brain metastatic CRC samples for expression experiments, we only involved FFPE samples of BM and primary tissues from two additional CRC patients and performed RNA-Seq in this revision. The standard STAR-featureCounts pipeline was used for the quantification of gene expression (See Methods). To characterize the expression pattern of DNA damage response genes in BMs and primary tissues, we performed differential expression analysis between brain metastasis and primary cancers followed by gene set enrichment analysis (GSEA) on lists ordered according to log fold change. Interestingly, we observed that several genes related to mismatch repair (MMR) or homologous recombination (HR) were downregulated in metastasis tissues (Supplementary Table 6). GSEA analysis also suggested that MMR and HR gene sets were significantly negatively enriched in brain metastasis tissues (Supplementary Figure 3). These results further support our findings at the DNA level. We added the description in the methods (Page 9, lines 171-188; Page 11, lines 218-224) and the results (Page 16, lines 335-341).

3. When the authors report that mutations were in the brain met only, and not in the primary tumor, how powered were they to detect that mutation in the primary? The authors have included power calculations but do not include this information. For example, the authors report that patients BM013 and BM014 carried BM-only mutations on *RAD51B* and *PAXIP1*. They need to give us power calculations for what the power was to detect those mutations in the primary tumor, in order to confirm they were absent. Similarly, they report that BM004 carried BM-only mutations in *POLB* gene. How powered were they to detect mutations in *POLB* in the primary

tumor These are not included in their power calculations in Supplementary Table 5. This is a critical part of this paper. When claiming BM only mutations, or primary tumor only mutations, we need to know how powered they were to detect these mutations in the other site where they were reported absent.

Response: We thank the reviewer for raising this important point. In this revision, we added power information of the mutations in both brain metastasis and primary tissues in Supplementary Table 7. To calculate the power of mutation detection in primary tissues of a CRC patient, we obtained the reference and alternative counts for each mutation identified in either the BM or primary tissues of the patient and estimated the cancer cell fraction and power by ABSOLUTE v1.2. All results were re-evaluated and no great difference was observed in the major findings.

4. A number of grammatical and linguistic errors need to be corrected.

Response: We apologize for the errors. In this revision, we have carefully reviewed the manuscript and proofed the grammatical and linguistic errors. The manuscript has also been edited by two native English speakers.

Reviewer #2 (Remarks to the Author):

I am glad to know the authors conducted whole-genome sequencing on “trios” tissues from 8 patients and provided some interesting results on signature analysis. Now this work includes 19 patients with brain metastatic CRC, 5 patients with liver metastatic CRC. Even though the manuscript has been much improved, I still have some concerns:

1. Chemo- and radiotherapy have been performed for the patients prior to brain metastatic in this study. The authors involved patients with liver metastasis to show that treatment didn't lead to great difference. I know it is difficult to obtain brain metastatic CRC samples before these treatment, but using liver metastasis cannot prove small difference in brain metastasis. Do the authors correct the mutations based on adjacent normal samples when you compare brain metastatic CRC samples with primary CRC samples? Normal cells may proliferate slower than tumor cells, so it could be less mutations in those.

Response: We agree with the reviewer. Although small difference was observed between the liver metastasis and primary tissues, the evidence that treatment didn't lead to great difference was still indirect. However, in this study, we cannot correct the mutations based on adjacent normal samples when comparing brain metastatic CRC samples with primary samples because brain metastases are commonly adjacent to the functional cortex¹ and adjacent normal samples of brain metastases are also difficult to obtain in a surgery.

In this revision, we found some additional clues from clonal evolution analysis. We inferred clonal evolution in 11 trios, which have WES data with relatively deep depth and involved multiple samples from each primary tissue. For each “trio”, we first estimated the clonal number based on the cancer cell fraction and constructed the phylogenetic tree based on a consensus model obtained from primary and brain metastasis (BM) samples. In this analysis, we successfully obtained consensus models in 7 of 11 “trios” and constructed phylogenetic trees accordingly. Interestingly, we identified 5 founding mutations of brain metastasis on DNA damage response genes (i.e., *BRCA2*, *POLB*, *CHEK1*, and *SLX4*) in 3 patients (Figure 2 and Supplementary Figure 6). The results indicated that these mutations appear at the founding clones of metastasis and may contribute to the subsequent signatures of DNA damage response deficiency in brain metastasis. We have added the methods (Page 13, lines 260-270) and the results (Page 18, lines 375-377 and lines 383-384; Page 19, lines 394-396) in this revision. We have also added the discussion of limitation. (Page 24, lines 525-528).

2. In the reference 5 cited, Naxerova et al. reconstructed the evolutionary relationship of primary tumors, lymph node metastases, and distant metastases. They found that in 65% of cases, lymphatic and distant metastases arose from independent subclones in the primary tumor, whereas in 35% of cases they shared common subclonal origin. Do you conduct similar analysis?

Response: In this revision, we have tried to infer clonal evolution in 11 trios, which have WES data with relatively deep depth and involved at most two samples from each primary tissue. Limited by the lack of multiple samples from tissues (BMs, primary tissues and normal), we cannot draw similar conclusion as Naxerova et al. have done. However, clonal analysis further strengthen and extended our results. We identified four BM founding mutations in DNA damage response genes (i.e., *BRCA2*, *POLB*, *CHEK1* and *SLX4*) (Figure 2 and Supplementary Figure 6). We also noticed that the two recurrent mutations in *SCN7A* (p.L805P/R) were also founding

mutations of BM samples (Supplementary Figure 6). These results indicated that these mutations should appear in the initial stage of brain metastasis and may contribute to the brain metastasis. We have added the methods (Page 13, lines 260-270) and the results (Page 18, lines 375-377 and lines 383-384; Page 19, lines 394-396; Pages 22, lines 459-461) in this revision.

3. The mutation patterns of different mutational signatures should be shown somewhere.

Response: In response to the reviewer's comments, we presented the mutation patterns of COSMIC mutational signatures in Supplementary Table 3. The mutation pattern of 4 different signatures was presented graphically in Figure 1C.

4. In figure 1A, C->T mutation in primary tumor is significantly higher than in brain metastatic. The authors cited reference 18, but I cannot find any description about C>T substitutions have been associated with epithelial tumors. In addition, C>T substitutions are very common in different mutational signatures. It looks it is not consistent with elevated mutational signatures in BM samples.

Response: The reviewer is correct. C>T substitutions are common in multiple mutational signatures (e.g., COSMIC signatures 1,6,7,11 etc.). Thus, the discussion here is inaccurate without considering the detailed mutational signatures. In the revision, we remove the unclear discussion and related reference 18 (Page 16, line 323).

5. It is not clear what are aging-related mutations on line 288. It may be a typo for "the primary metastatic CRC samples".

Response: We apologize for the unclear description. Aging-related mutations indicate the mutations with an aging-related mutational signature. For clarification, we have revised the description as follows (Page 16, lines 331-333):

"Interestingly, the proportion of the aging-related mutational signature was almost 80% in our cohort of the primary metastatic CRC samples but less than 50% in the TCGA CRC samples."

6. On line 323, the authors found that TP53 was the genes with the highest mutation rate, but almost all TP53 mutations were shared between BM and primary tissues, suggesting that TP53 mutations are drivers and occur prior to BM. Do the authors check TP53 mutations in normal tissue samples?

Response: We agree with the reviewer. In our clonal evolution analysis in this revision, we found that all *TP53* mutations appeared in the founding clone of primary CRC tissues (Supplementary Figure 6). Additionally, we tried to call somatic mutations in adjacent normal tissues of primary CRC tissues using the Mutect tumor-only mode and annotated the results using gnomad, ExAC and custom germline databases. However, no evident mutations remained after quality check and germline filtering, possibly due to the low cell fraction of mutations in normal tissue samples. In this revision, we have added the description of subclonal analysis of *TP53* mutations in Page 18, lines 374-375.

7. The authors have shown comprehensive analysis in nucleotide level, also expression level for some of genes. They also show most of potential driver genes related to BM are missense mutations, for example, *IKZF1*. In figure 4B, it has been shown in TCGA data, the expression of

IKZF1 is significantly associated with poor survival of CRC patients. I would ask whether IKZF1 is expressed in normal tissues or other cancers. It would be interesting to see how IKZF1 is related to poor survival of CRC patients.

Response: The reviewer raised an interesting question on *IKZF1* in colorectal cancer metastasis. *IKZF1* is a hematologic system specific expressed gene (Referee Figure 1) and was reported to play a critical role in lymphocyte differentiation and development^{2,3}. Recently, the MKSCC study also reported that *IKZF1* is a significantly recurrent mutated gene in MSS CRC cancer, indicating the importance of *IKZF1* in colorectal cancer⁴. However, we found that *IKZF1* was downregulated in TCGA CRC tumor tissues compared with adjacent normal tissues (Referee Figure 2A). To further determine the origin of *IKZF1* expression in bulk TCGA data, we included single-cell RNA-Seq data of CRC from a recent study⁵. Interestingly, though *IKZF1* was mainly expressed in lymphocytes and was downregulated in CRC tumor tissues, single cell RNA-Seq data suggested that it was also highly expressed in 4.0% of tumor cells (FPKM >5) and in none of the normal mucosa cells ($P=0.009$, Referee Figure 2B). The results suggested that *IKZF1* is expressed in both tumor cells and lymphocytes. Thus, the mechanisms of the *IKZF1* role in CRC prognosis could be complicated and may involve tumor cells as well as infiltrating lymphocytes. A series of well-designed experiments are required to further investigate the underlying mechanism. In this study, we presented the analysis of single-cell RNA-Seq data in Supplementary Figure 12 and added a discussion on Page 21 lines 438-450.

Referee Figure 1. The expression of *IKZF1* in normal tissues. *IKZF1* is a hematologic system specific expressed gene.

Referee Figure 2. A. *IKZF1* is down-regulated in TCGA CRC bulk tissues. B. *IKZF1* is highly expressed in 4.0% of CRC tumor cells and in none of the normal mucosa cells

8. I am interested in how authors generated supplementary Figure 2. The brief method should be introduced in Methods or Figure legends, including other figures. What's the unit of signature 17 means in x axis?

Response: We apologize for the unclear Supplementary Figure 2. In this figure, the x axis is the proportion of signature 17 in all samples. We inferred the proportion of signature 17 by R package deconstructSigs according to the pattern of signature 17 reported by COSMIC. The mutation burden was defined as the number of somatic mutations per megabase (/MB) of genome examined. In this revision, the Supplementary Figure 2 has been moved to Supplementary Figure 4 and we have added necessary methods and figure legends in Supplementary Figure 4 as well as in other figures.

9. On line 203, full name of CNA should be introduced.

Response: We apologize for unclear abbreviation. In this revision, we provided the expanded form of CNA at its first appearance (Page 12, line 238).

References:

1. Sekhar LN, Fessler RG. *Atlas of Neurosurgical Techniques: Brain*. Thieme (2016).
2. Cytlak U, *et al.* Ikaros family zinc finger 1 regulates dendritic cell development and function in humans. *Nature communications* **9**, 1239 (2018).
3. Oliveira VC, Sodre ACP, Gomes CP, Moretti NS, Pesquero JB, Popi AF. Alteration in Ikaros expression promotes B-1 cell differentiation into phagocytes. *Immunobiology* **223**, 252-257 (2018).
4. Yaeger R, *et al.* Clinical Sequencing Defines the Genomic Landscape of Metastatic Colorectal Cancer. *Cancer cell* **33**, 125-136 e123 (2018).
5. Li H, *et al.* Reference component analysis of single-cell transcriptomes elucidates cellular heterogeneity in human colorectal tumors. *Nature genetics* **49**, 708-718 (2017).

Reviewer #3 (Remarks to the Author):

It is commendable that the authors have increased the sample number to compare liver metastasis samples. The manuscript could still be strengthened. It would be beneficial to graphically present mutational signatures for each tumor, not just the numerical values that are presented in the new supplemental data. Figure 1 image and legend are still not clear.

Response: Thank you for the encouragement of the reviewer. In response to the reviewer's comments, we have presented the mutational signatures for each tumor in Supplementary Figure 2. We also have added necessary figure legends of Figure 1 to make it clearer.

More clinical data should be included in the methods section. It is especially necessary to know which therapies have been given prior to surgery.

Response: In response to the reviewer's comments, we have added the specific chemotherapy prior to brain surgery of 19 CRC patients with brain metastases in Supplementary Table 1 and the chemotherapy of 5 CRC patients with liver metastases prior to liver surgery have been added in the Supplementary Table 2.

One of the main conclusions is that DNA damage signatures increased in metastatic brain cancer, however this may simply be due to therapy exposure. Supplementary table 1 could be substantially improved by including the type of chemotherapy given as well as the additional liver metastasis samples included in the revised version.

Response: We thank the reviewer for raising this important point. In previous revised version, we included tissues from a total of 10 patients with liver metastasis (liver metastasis, colorectal primary and adjacent normal) tissues to prove that the treatment before metastasis may not lead to the great difference observed in the study. In this revision, we have further supplemented the specific chemotherapy of each patient in the Supplementary Table 1 and 2 in response to the reviewer's comments. Most BM patients (13/19) accepted the same chemotherapy regimens with liver metastatic patients before the surgery of metastases. Thus, we compared the proportion of mutational signatures between BM and primary tissues after excluding the 6 patients who accepted additional treatments and the results were consistent with Figure 1C (Referee Figure 3).

Referee Figure 3. We observed a consistent difference between BM and primary tissues after excluding the 6 patients who accepted additional treatments.

It has also come to my attention that the majority of brain metastases studied (14/19) are from rectal primary cancers. Radiotherapy is usually given in the setting of rectal cancer. Only a single case is from the distal colon, which is the most common site of primary colorectal cancer. My

concern is that this series is not representative of most colorectal cancers. Are rectal cancers known to more commonly metastasize to the brain? This may be reasonable, however should be highlighted in the manuscript.

Response: To answer the question, we searched the database of PubMed and Google Scholar using the key words “(rectal or rectum) and (cancer or neoplasm or tumor) and (Brain Metastases)”. Next, we found 20 relevant articles and 18 articles reported that more patients with rectal cancers have the tendency to metastasize to the brain (1-18, Referee Table 1). Only 2 articles showed that the percentage of colon cancer was higher than that of rectal cancer in patients with BM (19-20, Referee Table 1). Christensen, T.D. et al. conducted a systematic review of the incidence and patient characteristics of BM from CRC and showed that BM is a late-stage phenomenon, and young age, rectal primary and lung metastases are associated with an increased risk of developing BM⁶. Analyzing the patients from two Australian databases including 5,967 patients revealed that rectal cancer was more frequently associated with brain (RR=1.7; p=0.002) and lung metastases (RR=2.0; p<0.001) compared with the remaining left colon primaries⁷. This evidence suggested that rectal cancers more commonly metastasize to the brain. We have discussed the point in Page 25 (lines 536-540).

Referee Table 1. The list of 20 relevant articles which mentioned the association between brain metastasis and anatomy classification of colon and rectal cancer.

	Year	Title	Journal
1	1996	Colorectal Carcinoma and Brain Metastasis: Distribution, Treatment, and Survival	Annals of Surgical Oncology
2	2004	Patients with brain metastases from gastrointestinal tract cancer treated with whole brain radiation therapy: Prognostic factors and survival	World Journal of Gastroenterology
3	2004	Brain metastasis from colorectal cancer	International Journal of Colorectal Disease
4	2009	Brain metastases from colorectal cancer: risk factors, incidence, and the possible role of chemokines	Clinical Colorectal Cancer
5	2009	Brain metastases in colorectal cancers	World Journal of Surgery
6	2010	High chromosomal instability in brain metastases of colorectal carcinoma	Cancer Genetics and Cytogenetics
7	2011	Brain metastases from colorectal carcinoma: prognostic factors and outcome	Journal of Neuro-Oncology
8	2012	Brain metastases from colorectal cancer: the role of surgical resection in selected patients	Colorectal Disease
9	2012	Brain metastasis from colorectal cancer: prognostic factors and survival	Journal of Surgical Oncology
10	2012	Clinical features and course of brain metastases in colorectal cancer: an experience from a single institution	Current Oncology
11	2014	The clinical and pathological features of 133 colorectal cancer patients with brain metastasis: a multicenter retrospective analysis of the Gastrointestinal Tumors Working Committee of the Turkish Oncology Group (TOG)	Medical Oncology

12	2014	Prognostic factors for patients with advanced colorectal cancer and symptomatic brain metastases	Clinical Colorectal Cancer
13	2014	Brain metastases from colorectal cancer: main clinical factors conditioning outcome	International Journal of Colorectal Disease
14	2015	Brain Metastasis in Colorectal Cancer Patients: Survival and Analysis of Prognostic Factors	Clinical Colorectal Cancer
15	2015	Stereotactic radiosurgery in the treatment of brain metastases from gastrointestinal primaries	Journal of Neuro-Oncology
16	2016	Clinical features and prognostic factors of brain metastasis from colorectal cancer	Zhonghua Zhong Liu Za Zhi
17	2017	Risk factors for brain metastases in patients with metastatic colorectal cancer	Acta Oncologica
18	2018	Brain metastasis from colorectal cancer: a single center experience	Annals of Surgical Treatment and Research
19	1996	Brain Metastases from Colorectal Carcinoma:The Long Term Survivors	Cancer
20	2008	Neurosurgical management and postoperative whole-brain radiotherapy for colorectal cancer patients with symptomatic brain metastases	Journal of Cancer Research and Clinical Oncology

There are some differences in therapy regimen between the patients, did this correlate with mutational signatures?

Response: To answer the reviewer's question, we compared the proportion of 4 mutational signatures in BM tissues with and without specific therapy (radiotherapy or targeted therapy). We observed that neither radiotherapy nor targeted therapy was significantly associated with the proportion of the signatures (Referee Figure 4).

Referee Figure 4. The comparison of the proportion of 4 mutational signatures in BM tissues with and without specific therapy (radiotherapy or targeted therapy).

References:

6. Christensen TD, Spindler KL, Palshof JA, Nielsen DL. Systematic review: brain metastases from colorectal cancer--Incidence and patient characteristics. *BMC Cancer* **16**, 260 (2016).
7. Prasanna T, *et al.* The survival outcome of patients with metastatic colorectal cancer based on the site of metastases and the impact of molecular markers and site of primary cancer on metastatic pattern. *Acta Oncologica* **57**, 1438-1444 (2018).

REVIEWERS' COMMENTS:

Reviewer #1 (Remarks to the Author):

The authors have submitted an improved manuscript.

Reviewer #2 (Remarks to the Author):

The authors conducted an analysis of whole-exome sequencing (WES) and whole-genome sequencing (WGS) data on 19 "trios" of patient-matched BMs, primary CRC tumors, and adjacent normal tissue. In this version, they conducted subclonality analysis and inferred clonal evolution, they successfully obtained consensus model in 7 of 11 "trios". They identified 5 founding mutations of brain metastasis on DNA damage response genes. The results show different conclusion with previous studies and were discussed properly by authors. They also analyzed single-cell RNA-seq data to determine the origin of IKZF1 gene expression. All in all, the authors try to understand the mechanisms of brain metastasis of colorectal cancer from genome-wide analysis, the methods they used are robust, and they have conducted strong analysis in multiple patient samples those are hard-to-obtain because of rare occurrence but lethal. The resulted data will provide a better understanding of the BM mutational landscape and for other metastasis studies. My concerns have been solved. The paper in written has been improved much better now, but the authors should continue to polish for a better reading, the authors should correct some problems in line 459 and Supplementary Figure 1.

Yingjie Zhu

University of Texas Medical Branch

Reviewer #3 (Remarks to the Author):

I am satisfied that the authors have addressed my concerns.

Vicki Whitehall

REVIEWERS' COMMENTS:

Reviewer #1 (Remarks to the Author):

The authors have submitted an improved manuscript.

Response: Thanks for the encouragement of the reviewer.

Reviewer #2 (Remarks to the Author):

The authors conducted an analysis of whole-exome sequencing (WES) and whole-genome sequencing (WGS) data on 19 “trios” of patient-matched BMs, primary CRC tumors, and adjacent normal tissue. In this version, they conducted subclonality analysis and inferred clonal evolution, they successfully obtained consensus model in 7 of 11 “trios”. They identified 5 founding mutations of brain metastasis on DNA damage response genes. The results show different conclusion with previous studies and were discussed properly by authors. They also analyzed single-cell RNA-seq data to determine the origin of IKZF1 gene expression. All in all, the authors try to understand the mechanisms of brain metastasis of colorectal cancer from genome-wide analysis, the methods they used are robust, and they have conducted strong analysis in multiple patient samples those are hard-to-obtain because of rare occurrence but lethal. The resulted data will provide a better understanding of the BM mutational landscape and for other metastasis studies. My concerns have been solved. The paper in written has been improved much better now, but the authors should continue to polish for a better reading, the authors should correct some problems in line 459 and Supplementary Figure 1.

Yingjie Zhu

University of Texas Medical Branch

Response: Thanks for the encouragement of Dr. Zhu. In this revision, we have further polished the manuscript and corrected the problems in line 459 (line 277-278 in this version) and Supplementary Figure 1.

Reviewer #3 (Remarks to the Author):

I am satisfied that the authors have addressed my concerns.

Vicki Whitehall

Response: Thanks for the approval of Dr. Whitehall.